# Manuka Honey Inhibits Human Breast Cancer Progression in Preclinical Models

**DOI:** 10.3390/nu16142369

**Published:** 2024-07-22

**Authors:** Diana C. Márquez-Garbán, Cristian D. Yanes, Gabriela Llarena, David Elashoff, Nalo Hamilton, Mary Hardy, Madhuri Wadehra, Susan A. McCloskey, Richard J. Pietras

**Affiliations:** 1Division of Hematology-Oncology, Department of Medicine, UCLA David Geffen School of Medicine, Los Angeles, CA 90095, USArpietras@mednet.ucla.edu (R.J.P.); 2UCLA Jonsson Comprehensive Cancer Center, Los Angeles, CA 90095, USA; delashoff@mednet.ucla.edu (D.E.); mary@maryhardy.com (M.H.);; 3Division of General Internal Medicine, Department of Medicine, UCLA David Geffen School of Medicine, Los Angeles, CA 90095, USA; 4School of Nursing, UCLA, Los Angeles, CA 90095, USA; 5Department of Pathology and Laboratory Medicine, UCLA David Geffen School of Medicine, Los Angeles, CA 90095, USA; 6Department of Radiation Oncology, UCLA David Geffen School of Medicine, Los Angeles, CA 90095, USA

**Keywords:** Manuka honey, breast cancer, estrogen receptor-positive breast cancer, triple-negative breast cancer, in vivo xenografts, AMP kinase signaling, mTOR, STAT3

## Abstract

Manuka honey (MH) exhibits potential antitumor activity in preclinical models of a number of human cancers. Treatment in vitro with MH at concentrations ranging from 0.3 to 5.0% (*w*/*v*) led to significant dose-dependent inhibition of proliferation of human breast cancer MCF-7 cells, but anti-proliferative effects of MH were less pronounced in MDA-MB-231 breast cancer cells. Effects of MH were also tested on non-malignant human mammary epithelial cells (HMECs) at 2.5% *w*/*v*, and it was found that MH reduced the proliferation of MCF-7 cells but not that of HMECs. Notably, the antitumor activity of MH was in the range of that exerted by treatment of MCF-7 cells with the antiestrogen tamoxifen. Further, MH treatment stimulated apoptosis of MCF-7 cells in vitro, with most cells exhibiting acute and significant levels of apoptosis that correlated with PARP activation. Additionally, the effects of MH induced the activation of AMPK and inhibition of AKT/mTOR downstream signaling. Treatment of MCF7 cells with increased concentrations of MH induced AMPK phosphorylation in a dose-dependent manner that was accompanied by inhibition of phosphorylation of AKT and mTOR downstream effector protein S6. In addition, MH reduced phosphorylated STAT3 levels in vitro, which may correlate with MH and AMPK-mediated anti-inflammatory properties. Further, in vivo, MH administered alone significantly inhibited the growth of established MCF-7 tumors in nude mice by 84%, resulting in an observable reduction in tumor volume. Our findings highlight the need for further research into the use of natural compounds, such as MH, for antitumor efficacy and potential chemoprevention and investigation of molecular pathways underlying these actions.

## 1. Introduction

Breast cancer (BC) is a leading cause of death in women worldwide. Approximately 60–70% of patients diagnosed with BC have estrogen receptor alpha (ERα) expression in their tumors [1,2,3]. Endocrine therapy has proven to be one of the most effective targeted treatments for BCs that express ERs and has accounted for significant improvement in progression-free and overall survival of BC patients over the past two decades [4,5]. However, a proportion of patients with early diagnoses, and essentially most patients with metastatic BC, ultimately become resistant to available endocrine therapies. In the absence of options to current treatments such as antiestrogens (tamoxifen, fulvestrant) or aromatase inhibitors (anastrazole, letrozole, exemestane) alone or combined with CDK4/6 inhibitors, cytotoxic chemotherapy is often the only alternative for clinical management. Similarly, chemotherapies are often used for patients with triple-negative breast cancer (TNBC), a subtype that occurs in about 15% of BC patients and cannot be managed with current endocrine or HER2-targeted therapies [6]. The emergence of endocrine resistance is one reason that BC is the first cause of cancer death in women around the world and in some racial/ethnic groups in the United States. Thus, the development of alternative treatments to help prevent the development of endocrine resistance and to improve long-term BC patient survival is urgently needed. 

Manuka honey (MH) is a monofloral honey that is produced from the nectar collected by honey bees (*Apis Mellifera*) when they pollinate the Manuka tea tree (*Leptospermum scoparium*). This honey was originally unique to New Zealand and some parts of Australia; however, it is now produced in many regions around the world. MH is known to exhibit antimicrobial, antioxidant, and tissue-protective/healing activities [7,8]. MH has an exclusive composition containing high concentrations of methylglyoxal, the main compound considered to be responsible for its antibacterial and antioxidant properties. Likewise, it is rich in flavonoids, phytochemicals, complex carbohydrates, vitamins, amino acids, and minerals [7,9,10,11]. While MH shares constituents with other types of honey, such as glucose oxidases, it also contains other specific phytochemical factors that may potentiate its biological activity [12]. Of note, natural phytoestrogens may be one such factor [13,14]. Whether the range of biological activities of honey is mediated by the same or different active compounds has not been fully studied. Historically, MH has been used in traditional medicine for wound healing, urinary and stomach symptoms, and to control fever. Extensive scientific and clinical evidence supports the utilization of honey for wounds, skin reactions, and damage to epithelial barriers following radiation treatment and/or chemotherapy [15,16,17]. In patients with chronic wounds, honey is reported to activate the innate immune system, inducing the migration of neutrophils and macrophages, stimulating angiogenesis, and preventing infection [15,16,17,18,19,20]. Emerging reports currently indicate that MH, being rich in polyphenols and flavonoids, also has notable anti-proliferative effects against several human cancer cells [16,18,21,22,23]. Of special note, independent reports provide additional evidence for the induction of apoptosis in cancer cells, including breast and colon cancer cells and melanoma, by MH treatment in vitro at concentrations as low as 0.6% (*w*/*v*). Moreover, results of recent investigations show that administration of MH in vivo exhibits significant anticancer activity when given alone and helps to prolong survival when used in combination with paclitaxel chemotherapy in preclinical mouse tumor models [16]. Mechanisms of antitumor action of MH and its constituent compounds are suggested to include activity as selective estrogen receptor modulators (SERMs), inhibitors of growth factor receptor signaling pathways, and blockade of the proliferation of breast cancer stem/progenitor cells, which play a critical role in tumor regeneration and spread following treatment with standard therapies in the clinic [16,24,25,26]. Accordingly, additional work is needed to define the benefits and mechanism of action of MH and its potential for use in the clinical management of individuals afflicted with specific types of cancers, particularly breast cancer.

In this study, we investigated the potential antitumoral effects of MH on estrogen receptor-positive and negative breast cancer. We found that MH and powder inhibit cell proliferation in a dose-dependent manner in vitro and in vivo. We further elucidated possible signaling pathways involved in MH mechanisms of action. Activation of AMP kinase (AMPK) and inhibition of downstream mTOR signaling, as well as STAT3, appear to be molecular targets that are mediating MH antitumor therapeutic action.

## 2. Materials and Methods

### 2.1. Reagents

Manuka honey MGO 550+ and dehydrated MH powder (MH cyclopower powder, a dehydrated MH complexed with natural cyclodextrins) were provided by Manuka Health New Zealand Limited (Auckland, New Zealand). Working stock solutions of the MH cyclopower powder were prepared by weighing aliquots and dissolving them in respective amounts of 15% ethanol/Hanks’ balanced salt solution buffer to give the desired concentrations. As a control for the sugar component of Manuka honey, we used 5% dextrose. As a control for another honey product, we used 5% Mesquite honey, a honey native to the southwestern United States and Mexico. Lastly, we used W6 Cavamax, an α-cyclodextrin powder, used in the manufacture of the Manuka honey cyclopower powder (Wacker BioSolutions, Ann Arbor, MI, USA). Honey and dextrose were dissolved in a cell culture medium as *w*/*v*.

### 2.2. Cell Lines

Human breast cancer cell lines were obtained from the American Type Culture Collection (ATCC) (Manassas, VA, USA) and cultured according to ATCC recommendations. ERα-positive human BC MCF-7 cells were cultured in DMEM or RPMI-1640 media, and triple-negative (ERα-/PR-/HER2-) human BC MDA-MB-231 cells were cultured in RPMI-1640. Culture media were supplemented with 10% fetal bovine serum (FBS; Gemini Bio-Products, Sacramento, CA, USA), 100 units/mL penicillin, 100 µg/mL streptomycin sulfate, and 2.5 µg/mL amphotericin B (Gemini Bio-Products, Sacramento, CA, USA) [27,28]. Human H2110 non-small cell lung cancer (NSCLC) and Panc1 pancreatic cancer cells were also obtained from the ATCC (Manassas, VA, USA)and were routinely maintained in RPMI 1640 medium with 10% FBS. Cell cultures were maintained at 37 °C in a 5% CO_2_ tissue culture incubator. A control non-malignant human mammary epithelial cell line (HMEC) was obtained from commercial sources (Invitrogen/ThermoFisher Scientific, Carlsbad, CA, USA) and maintained in vitro as per the supplier’s recommendation [29].

### 2.3. Cell Proliferation Assay

Malignant and non-malignant control cells were seeded in 96-well plates at 2–3 × 10^5^ cells/well in complete medium. After 48 h, cells were treated with the indicated concentrations of respective treatments for 72 h in phenol-red free RPMI-1640 media. Treatments were performed in quadruplicate, and experiments were repeated at least three times. Cell number counts were performed, and cell proliferation was quantitated by colorimetric assays using the CellTiter 96^®^ Aqueous (Promega, San Luis Obispo, CA, USA) assay as per the manufacturer’s instructions [30]. The latter assay quantifies the amount of metabolically viable cells by determining the activity of NADH-dependent cellular oxidoreductase enzymes. The data are generally presented as percent cell viability for the treatment groups as compared to that of control untreated cells. 

### 2.4. Western Blot Analysis

MCF-7 cells were plated in RPMI-1640 with 10% FBS. After 24 h, cells were incubated in the presence of 0.6%, 2.5%, or 5.0% MH for 24 h in phenol-red free medium with 5% FBS. Cell lysates were prepared according to established protocols using RIPA buffer, and the protein concentration was determined using the BCA Protein Assay Kit (PIERCE/ThermoFisher Scientific, Carlsbad, CA, USA) [29,30,31]. Forty micrograms of total cell protein were then resolved by 4–15% SDS-PAGE and transferred to a PVDF membrane. Membranes were blocked in TBS-T (Tris-buffered saline, 0.1% Tween 20) with 5% non-fat dry milk or 5% BSA. Primary antibodies were from Cell Signaling Technology (Beverly, MA, USA): (ADP-Ribose) polymerase (PARP) (#9542), STAT3 (#12640), pAktSer473 (#9271), Akt #9272, pAMPKThr172 (#2535), AMPK #2532, pS6 ribosomal protein ser235/236 (#2211), S6 (#2217), and Active Motif anti-Ser727-phosphorylated STAT3 cat# 39613, dil 1:1000. In these studies, Actin (dil 1:1000, cat# PA5-58528, Invitrogen/ThermoFisher Scientific, Carlsbad, CA, USA) and GAPDH (cat# 12004167, BIORAD, Hercules, CA, USA) were used as a loading control.

### 2.5. Annexin V Staining Assay

MCF7 and MDA-MB-231 BC cells were cultured in 6-well plates at a concentration of 250,000 cells/well for 24 h. The next day, cells were treated with either 2.5% MH, 5.0% MH, 5.0% dextrose, 5.0% Mesquite honey, or 10 µM Camptothecin for 24 and 48 h. Cells were collected with Accutase™ cell dissociation reagent, washed and stained using an Annexin V-FITC and 7-AAD Dead Cell Apoptosis Detection Kit (R&D Systems), and analyzed using flow cytometry. 

### 2.6. Human Tumor Xenografts in Nude Mice

Animals were housed in a pathogen-free environment with controlled light and humidity and received food and water ad libitum. All studies were approved by the UCLA Animal Research Protection Committee. Human MCF-7 cells were inoculated subcutaneously at 2 × 10^7^ cells/animal into the dorsal area of 6-week-old female athymic nude mice (Charles River Laboratories) primed with 0.36 mg s.c. estradiol-17β (E2β) in a biodegradable binder 60-day release pellet (Innovative Research of America, Sarasota, FL, USA) beginning 3 days prior to cell inoculation as before [27,30]. Treatment was initiated when tumors grew to 50–80 mm^3^. Animals were randomized by weight and tumor size at the start of the experiment, with 5–7 animals included in each treatment group. Treatments included MH or control vehicle administered by oral gavage. Oral gavage (0.2 mL volume) with 50% *w/v* Manuka honey or control (50% dextrose) was performed twice daily from days 1 to 14, then daily thereafter to day 42. The tumors were measured using calipers on the days indicated, and tumor volume was calculated by (*l* × *w* × *w)*/2, with tumor length *l* and tumor width *w* in mm. Data were presented as the mean ± SEM for tumor volumes measured in cubic mm. Data were analyzed by use of ANOVA and Student’s *t*-test statistical approaches as before [29,30,31,32]. 

### 2.7. Statistics

Statistical differences regarding in vitro cell proliferation and/or apoptosis assays were analyzed using Student’s *t*-test. ANOVA was used for comparison of tumor xenograft volumes in preclinical models. All results were expressed as mean ± SEM, with *p* < 0.05 considered as statistically significantly different.

## 3. Results

### 3.1. Manuka Honey Reduces the Proliferation of Human MCF-7 Breast Cancer Cells In Vitro

We first investigated the effect of in vitro treatment with varying concentrations of MH (0.3–5.0%) or a dehydrated MH powder for 72 h (Figure 1A). Using ERα-positive MCF-7 BC cells, we observed a significant dose-dependent inhibition of cell proliferation among cells treated with MH as compared to cells treated with vehicle control (*p* < 0.01). In contrast, MH treatment elicited a more modest antitumor effect on triple-negative BC cells MDA-MB-231 (ER-/PR-/HER2-) (*p* < 0.05). Similar differences in the antitumor responses among MCF-7 and MDA-MB-231 cells were also evident when tumor cells were treated with Manuka powder (Figure 1B), with only MCF-7 cells exhibiting a significant dose-dependent suppression of proliferation in vitro (*p* < 0.01). Of note, Manuka honey at 2.5–5.0% (*w*/*v*) also provoked a significant 75% reduction (*p* < 0.05) in the proliferation of human H2110 non-small cell lung cancer and PANC1 pancreatic cancer cells that express the aromatase enzyme needed for local estrogen biosynthesis as well as estrogen receptors (Appendix A) [33]. 

Tamoxifen is widely used in the clinic as an antiestrogen therapy, with evidence showing a significant reduction in BC mortality in ER-positive early BC. Tamoxifen competes with estradiol for binding to the ER and serves as a selective estrogen receptor modulator. Although effective, tamoxifen has important drawbacks, notably a limited period of activity before the emergence of drug resistance. Accordingly, we assessed the effects of 2.5% MH alone and combined with 10 µM tamoxifen (TM) among MCF-7 BC cells. Non-malignant human mammary epithelial cells (HMECs) were also investigated in parallel to test the potential toxicity of MH as compared to TM and with their combination effect. When treating with MH, we found that MCF-7 cells exhibited a clear inhibition of proliferation (*p* < 0.01), while HMEC proliferation was not affected (Figure 1C). As expected, TM significantly reduced the proliferation of MCF-7 cells (*p* < 0.01), with only a modest inhibitory effect on HMECs. However, when combination therapy with MH + TM was tested, we found that MCF-7 cell proliferation was markedly suppressed and was significantly less than that of either treatment administered alone (*p* < 0.001). 

### 3.2. MH Induces Apoptosis in MCF-7 Breast Cancer Cells In Vitro

In the next set of experiments, we investigated a potential mechanism by which Manuka honey was disrupting cancer cell progression. Thus, one of the earliest events in the process of apoptosis is the loss of cell membrane asymmetry as detected by Annexin V staining. MCF-7 and MDA-MB-231 cells were harvested at 24 or 48 h after treatment with control vehicle, different concentrations of Manuka honey, Mesquite honey, dextrose, and camptothecin, stained with Annexin V-FITC and 7-AAD, and then subjected to flow cytometric analysis. Notably, camptothecin is an anticancer chemotherapy known to induce apoptosis and is used as a positive control [34]. As can be seen in Figure 2A, there was a rapid and dose-dependent increase in the number of MCF-7 cells undergoing apoptosis (Annexin V-positive), particularly late apoptosis (Annexin V-positive and 7-AAD-positive) after culture with increasing concentrations of Manuka honey at 24 or 48 h (Figure 2A) (*p* < 0.05). In addition, MDA-MB-231 cells exhibited evidence of apoptosis but to a lesser extent than MCF-7 cells (Appendix A). 

Cells treated with camptothecin exhibit a similar pattern of response, with an apparent time-dependent increase in the number of apoptotic cells. In contrast, MCF-7 cells treated with 5% (*w*/*v*) dextrose or 5% (*w*/*v*) Mesquite honey did not exhibit comparable increments in the number of apoptotic cells after 48 h treatment as compared to Manuka honey-treated cells (Figure 2A). 

These findings suggest that the pro-apoptotic effects of Manuka honey are not associated with the sugar content of Manuka, nor are the pro-apoptotic effects associated with Mesquite honey from a different region. 

### 3.3. Manuka Honey Induces PARP-Cleavage Leading to Tumor Cell Apoptosis

Overall, these findings suggest that the death of breast cancer cells following exposure to relatively low concentrations of Manuka honey occurs via an apoptotic mechanism. A critical step in the apoptosis pathway is the activation of caspases that lead, in turn, to the cleavage of several cell substrates required for cell viability [35]. One target protein is the DNA repair enzyme poly(ADP-ribose) polymerase (PARP). Thus, we investigated the effect of Manuka honey treatment on PARP cleavage using a monoclonal antibody against PARP that detects the full length (116 kD) and the cleaved (89 kD) forms of PARP (Figure 2B). Lysates of MCF-7 cells were prepared following treatment with Manuka honey or control for 48 h and then subjected to immunoblot analysis with a PARP-specific antibody. Cleavage of PARP into an 89 kD fragment was evident, with maximal activity at 5% (*w*/*v*) MH. Thus, MH appears to effectively induce the caspase pathway, leading to apoptosis of MCF-7 cells.

### 3.4. Manuka Honey Activates AMPK and Inhibits Downstream PI3K/AKT/mTOR Signaling

Activation of AMP-activated protein kinase (AMPK) has been found to inhibit tumor cell progression in several cancer types [36]. This effect is mediated in part by inhibition of the downstream phosphoinositide 3-kinase (PI3K) AKT/mammalian target of rapamycin (mtTOR) signaling pathway. Blockade of PI3K/AKT/mTOR signaling has been shown to be effective in overcoming resistance in ER+ BC [37]. Antioxidant properties of Manuka honey have been attributed to its uniquely high content of polyphenols such as caffeic acid, caffeic acid phenethyl ester (CAPE), chrisin, etc., known to activate AMPK and inhibit the PI3K/AKT/mTOR pathways [38,39,40]. In order to determine if MH had any effect on AMPK activation, we treated MCF7 cells for 24 h with increasing concentrations of MH (0–5%) and used as a control 5% Mesquite honey. Phosphorylation of AMPK (Thr172) was detected in a dose-dependent manner, with activity detected with as low as 0.6% MH. AMPK activation was accompanied by inhibition of phosphorylation of AKT and mTOR downstream signaling protein S6 that was specific for MH and not Mesquite honey (Figure 3A). These results corroborate previous findings demonstrating different polyphenols can activate AMPK and inhibit PI3K/AKT/mTOR signaling in cancer.

### 3.5. Manuka Honey Inhibits STAT-3 Phosphorylation

Enhanced levels of tyrosine-phosphorylated signal transducer and activator of transcription 3 (*p*-STAT3) are found constitutively in about half of human breast cancers. These activated forms of STAT3 act, in turn, as oncogenic transcription factors to promote tumor progression. Phosphorylation of STAT3 is also associated with pro-inflammatory gene regulation [41]. To assess the effects of Manuka honey, we treated MCF-7 cells for 24 h with control or Manuka honey at 0.6%, 1.25%, 2.5%, or 5.0% (*w*/*v*). The results show a dose-dependent suppression of *p*-STAT3 (Ser727) levels, with maximal activity at 5.0% (*w*/*v*) MH (Figure 3B). These findings are consistent with earlier reports demonstrating that MH inhibits *p*-STAT3 in breast cancer cells [35].

### 3.6. Antitumor Activity of Manuka Honey in Human Breast Cancer Xenografts In Vivo

Given the significant antitumor effect of MH on human breast cancer cells in vitro, we next investigated the activity of Manuka honey using an animal tumor model in vivo [27,42]. Ovariectomized nude mice with estradiol supplements were implanted with ER-positive MCF-7 tumor cells subcutaneously in the flanks and treated with MH or control dextrose administered by oral gavage after tumors achieved a size of 50–75 cm^3^. Oral gavage (0.1 mL volume) with 50% (*w*/*v*) Manuka honey or dextrose control was performed twice daily from days 1 to 28 and then once daily thereafter to day 42. Tumor volume and animal survival were then followed (Figure 4).

Treatment with MH administered orally elicited a significant suppression of MCF-7 xenograft progression as compared to controls (*p* < 0.01). As shown in Figure 4, tumor growth in control-treated mice exhibited a progressive and sustained increase over the course of the experiment, while mice treated with MH exhibited a significant reduction in tumor progression and in tumor volume (Figure 4).

## 4. Discussion

Independent reports have provided evidence that honey such as Manuka honey exerts anti-proliferative effects against several types of cancer cells in vitro [16,23,24,25]. However, potential mechanisms for such anticancer action, particularly in vivo, remain to be fully elucidated. The current study investigated the effect of Manuka honey on the growth and progression of human breast cancer cells, using both in vitro and in vivo approaches. Our findings show that treatment of human breast cancer cells with MH leads to significant inhibition of tumor cell proliferation and the induction of apoptosis in vitro, while orally administered Manuka honey demonstrated significant activity as an anticancer or chemopreventive agent in vivo. Two representative breast cancer cell lines, ER-positive MCF-7 cells and triple-negative MDA-MB-231 cells, were selected for these studies. The results of in vitro experiments demonstrate that exposure to MH significantly suppresses proliferation in a dose-dependent manner in MCF-7 cells, while anti-proliferative effects in TNBC MDA-MB-231 cells are more limited. Additionally, we determined that MH treatment did not alter the in vitro proliferation of non-malignant human mammary epithelial cells (HMECs), suggesting less generalized toxic effects in normal cells. Of special note, MH treatment of ER-positive MCF-7 cells enhances the antitumor action of tamoxifen when MH is combined with the antiestrogen commonly used in breast cancer therapy in the clinic [1,5]. This result is consistent with other recent reports showing that Tualang honey promotes the anticancer activity induced by hydroxytamoxifen in MCF-7 cells [26,27]. It is further reported that certain phenolic compounds that are constituents of honey are phytoestrogens with structural similarity to mammalian estrogens and can potentially bind to estrogen receptors [28,33,42]. Hence, the occurrence of natural phytoestrogens as constituents of honey may be one explanation for these antitumor actions [13,14,26] and suggests that endocrine-sensitive breast cancer cells may be a reasonable target to explore for MH or MH derivatives.

Notably, MH stimulates apoptosis of MCF-7 cells in vitro, and this MH-induced apoptotic action correlates with the induction of PARP cleavage. However, apoptosis of MCF-7 cells was not observed after treatment with either equivalent concentrations of a regional US Mesquite honey or dextrose, the sugar that constitutes about 31% of honey [7,8,10,43]. Fernandez-Cabezudo et al. [16] reported previously that MH exerts its antitumor effects through apoptosis by activation of caspase 9 followed by caspase 3 and downregulation of expression of Bcl-2. Likewise, Tualang honey has been demonstrated to induce apoptosis in MCF-7 breast cancer cells through activation of caspases 3/7 and 9, suggesting a mitochondrial apoptosis pathway [27]. Importantly, treatment with Tualang honey was specific for malignant breast cancer cells since honey had no cytotoxic effect in MCF-10A normal mammary breast cells. Although detailed analyses of the effect of other types of honey on cancer cells remain to be achieved, our limited results suggest that differences in the antitumor activity of honey from different regions may potentially be due to variations in honey constituents, particularly in polyphenols and phenolic acids with known antitumor activities [13,16]. Together, these data suggest that honey exhibits anticancer effects due in part to its antiestrogen activity but also its ability to induce apoptosis through different mechanisms.

The current findings further provide evidence that orally administered MH has antitumor activity in stopping the progression of human breast tumor xenografts implanted in nude mouse models in vivo. Based on promising results from our in vitro studies, the antitumor effects of MH were tested using ER-positive MCF-7 tumor preclinical models to assess the antitumor effects of MH as compared to control treatments over a 42-day period. Overall, MH administered by oral gavage significantly inhibited the growth and progression of established human breast tumor xenografts in nude mouse models by 84%. In independent work, Fernandez-Cabezudo et al. [16] used an in vivo syngeneic mouse melanoma model to assess the antitumor effect of intravenously administered Manuka honey, alone or combined with paclitaxel chemotherapy, on the growth of established melanomas. Treatment with Manuka honey alone resulted in an approximate 33% inhibition of tumor growth, but greater control of tumor growth was observed in animals treated with paclitaxel in combination with Manuka honey, as well as a marked improvement in host survival in the dual treatment group. Ahmed et al. [13] further investigated the antitumor effects of Malaysian Tualang honey (TH) and Australian/New Zealand Manuka honey (MH) against carcinogen-induced breast cancer in rats. Treatment with orally administered honey at 1.0 gm/kg body weight/day for 120 days began when developing tumors achieved 10–12 mm in size. Animals treated with honey had a significantly slower rate of tumor growth as well as lower median tumor sizes and numbers of tumors as compared with control-treated groups. Treatment with honey also increased the expression of pro-apoptotic proteins and decreased the expression of anti-apoptotic proteins. Importantly, the findings of this study further showed that MH elicited a reduction in serum estradiol levels and a decrease in ERα in tumors as compared to controls. These findings are notable because postmenopausal women with elevated serum sex steroids, particularly estrogens, have an increased risk of breast cancer [44]. Estradiol binds and activates tumor cell estrogen receptors that act to promote proliferation and suppress apoptosis by both direct and indirect modulation of target gene transcription [1,2,3]. Accordingly, BC treatment with antiestrogens and/or aromatase inhibitors that reduce circulating estrogen levels is critical to manage disease progression in the clinic. Thus, MH and TH may inhibit ER-positive tumor progression in part by reducing circulating estrogen levels and potentially acting as selective estrogen receptor modulators to disrupt estrogen signaling pathways [13,14,28,33]. 

We note that prior studies similarly reported on varying levels of antitumor activity of honey as well as anti-metastatic activities against a number of tumor cell lines [13,21,23,25,34]. Notably, high levels of tyrosine-phosphorylated signal transducer and activator of transcription 3 (*p*-STAT3) are found to be constitutively activated in a number of malignancies, including almost half of all human breast cancers and act as oncogenic transcription factors. Thus, the current and previous findings that MH inhibits the phosphorylation of STAT3 in breast cancer cells may also play a role in the blockade of tumor progression [35]. Further, another mechanism we found contributing to the effects of MH is through activation of AMPK and inhibition of the AKT/mTOR downstream pathway. Polyphenols such as caffeic acid and derivatives have been shown to inhibit the growth of colon cancer cells [45] and reverse doxorubicin resistance in BC cells [46] via the AMPK/AKT/mTOR pathway modulation. Pinocembrin, a flavonoid that is rich in MH, inhibited BC cell proliferation and metastasis also through inhibition of the PI3K/AKT pathway [47]. Many properties of honey that have been described to aid in the process of antimicrobial and wound healing activity—such as activating the innate immune system, inducing the migration of neutrophils and macrophages, stimulating angiogenesis, and enhancing antibody production [13,17,18,25], may also serve to promote antitumor actions in vivo. While Manuka honey shares constituents (such as glucose oxidases) with other types of honey, it also contains other specific phytochemical factors that potentiate its biological activity, such as methylglyoxal [9,10,11,12]. As proposed by others [48,49], we concur that significant evidence suggests that the health benefits of fruits, vegetables, whole grains, and other plant foods can be attributed to the synergy or interactions of bioactive compounds and other nutrients in whole foods. Consequently, efforts to dissect specific chemical constituents with a given biological effect may prove futile. However, further investigation is needed to better understand which constituents of MH may underlie its antitumor activity and aid in the development of new anticancer and/or chemopreventive drugs. 

A major concern for many current anticancer drugs is their potential toxicity. Considerable efforts are being exerted to identify naturally occurring compounds, or their principal active constituents, with the potential to complement existing cancer therapeutic and/or chemopreventive modalities [50,51,52]. Prior independent reports have established that MH at doses similar to those utilized in this work causes no apparent systemic side effects as determined by comprehensive analyses of hematologic and clinical chemistry parameters to probe for alterations in cellular constituents of blood or chemical markers of organ dysfunction [13,24,25,43]. 

## 5. Conclusions

Our findings confirm that MH has potent anticancer properties through different mechanisms. MH’s unique composition, including phenolic compounds and methylglyoxal, has been reported to have antioxidant, antiseptic, and anticancer properties. MH inhibited in vitro cell proliferation of MCF7 cells in a dose-dependent manner and induced apoptosis through PARP activation. Further, MH activated AMPK and inhibited mTOR downstream signaling as well as STAT3. Notably, orally administered MH inhibited the growth of MCF7 tumor xenografts in vivo without major side effects. These findings indicate that natural compounds such as Manuka honey, with significant antitumor activity and selectivity towards hormone receptor-positive breast cancers, may be further developed as a supplement or potential alternative to cytotoxic anticancer drugs that have more non-selective adverse effects. 

## Figures and Tables

**Figure 1 nutrients-16-02369-f001:**
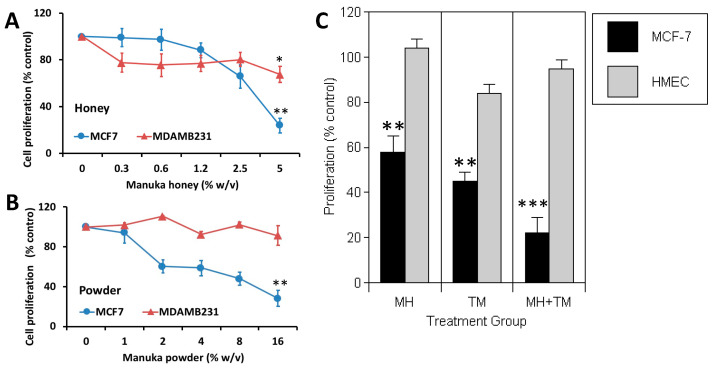
Manuka honey reduces the proliferation of ER-positive human breast cancer cells in vitro. ER-positive MCF-7 breast cancer cells and TNBC MDA-MB-231 cells were incubated in the presence of increasing concentrations of either (**A**) Manuka honey at 0.0 to 5.0% (*w*/*v*) or (**B**) Manuka powder at 0.0–16%. After 72 h, cell counts were performed using an MTS assay and by manual cell counts. The figures show tumor cell proliferation expressed as the mean percentage of vehicle-treated controls with SEM. Experiments were performed at least three times in independent experiments. (**C**) Manuka honey reduces the proliferation of MCF-7 cancer cells but not that of non-malignant mammary cells in vitro and enhances the antitumor action of the antiestrogen tamoxifen. Human MCF-7 tumor cells and non-malignant HMECs were cultured in vitro with 2.5% *w/v* Manuka honey (MH), 10 μM tamoxifen (TM), or both agents combined for 48 h. Cell proliferation was then quantitated and expressed as a percentage of that recorded in appropriate vehicle-treated controls. A higher 5% *w/v* MH concentration was also tested without a significant effect on HMEC proliferation. * *p* < 0.05, ** *p* < 0.01, *** *p* < 0.01, n > 3.

**Figure 2 nutrients-16-02369-f002:**
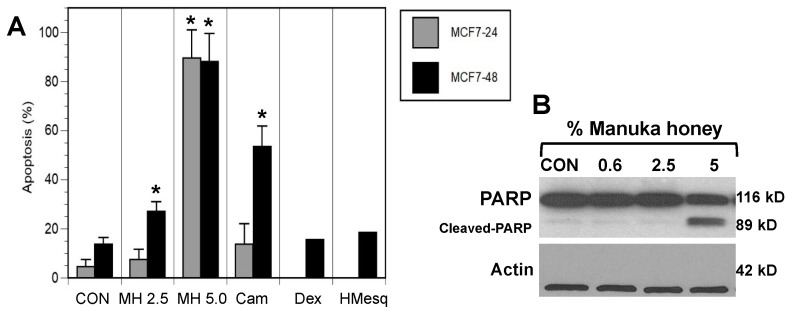
Induction of apoptosis of breast cancer cells by Manuka honey. (**A**) MCF-7 cells were treated with vehicle control (CON), 2.5% (*w*/*v*) (MH 2.5) or 5% (*w*/*v*) (MH 5.0) Manuka honey, camptothecin 1 μM (Cam), 5% (*w*/*v*) (Dex) dextrose, or 5% (*w*/*v*) (HMesq) Mesquite honey. After 24 and/or 48 h, cells were harvested and stained with Annexin V and 7-AAD to assess early and late apoptosis. Treatments with Manuka honey, particularly at 5.0% (*w*/*v*), elicited significant increments in apoptotic cells as compared to controls (* *p* < 0.05). Camptothecin, a positive control drug, elicited a similar increase in late apoptotic cells after 48 h, while treatment of MCF-7 cells with dextrose or Mesquite honey did not exhibit comparable increments in the numbers of apoptotic cells. (**B**) Treatment of MCF-7 cells with Manuka honey elicits increased poly (ADP-ribose) polymerase (PARP) cleavage. Cells were treated in vitro for 48 h with either control vehicle or Manuka honey at 0.6, 2.5, or 5.0% (*w*/*v*). Camptothecin was also used as a positive control.

**Figure 3 nutrients-16-02369-f003:**
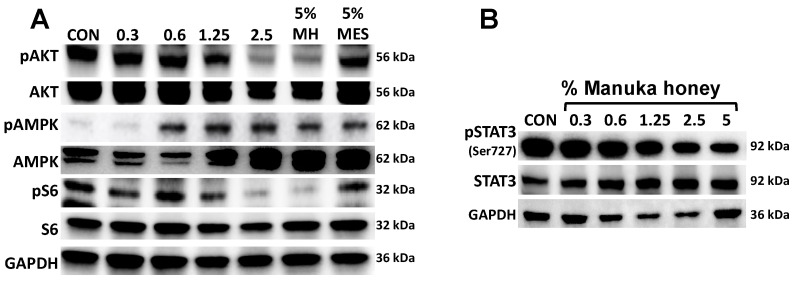
Manuka honey activates AMPK signaling and inhibits mTOR and STAT3 downstream signaling. (**A**) MCF7 cells were treated with increasing concentrations of Manuka honey (0–5%) and 5% Mesquite honey as control. After 24 h, cells were lysed and immunoblotted with specific antibodies. (**B**) MCF7 cells were treated with increasing concentrations of Manuka honey 0.3–5% (*w*/*v*). After 24 h, cells were lysed, and whole cell extracts were resolved by PAGE and immunoblotted with specific antibodies.

**Figure 4 nutrients-16-02369-f004:**
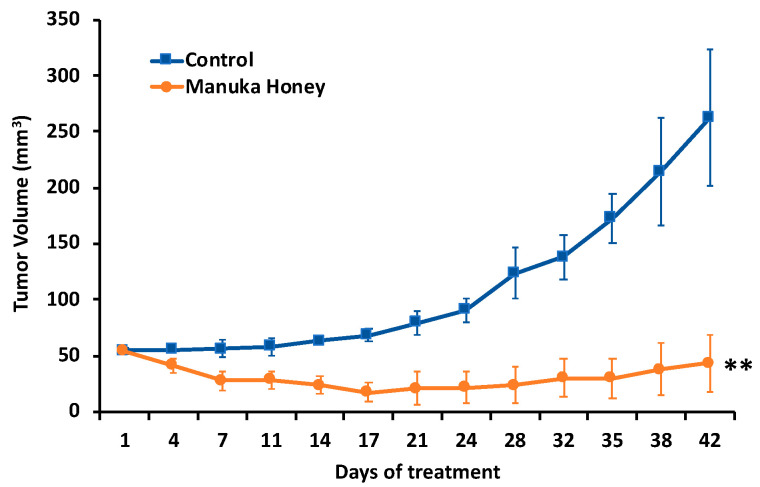
Antitumor activity of Manuka honey in human breast cancer xenografts in vivo. Ovariectomized nude mice (nu^−^/nu^−^, Charles Rivers) with estradiol supplements were implanted with MCF-7 tumor xenografts SQ and treated with Manuka honey or control administered by oral gavage after tumors achieved a size of 50–75 cm^3^. Oral gavage (0.1 mL volume) with 50% (*w*/*v*) Manuka honey or control was performed twice daily from days 1 to 14, then once daily thereafter to day 42. Treatment with Manuka honey administered orally elicited a significant suppression of MCF-7 xenograft progression as compared to controls (** *p* < 0.01) n = 5–7 mice per group.

## Data Availability

The original contributions presented in the study are included in the article and Appendix A; further inquiries can be directed to the corresponding authors.

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
