# Peer review of "Manuka Honey Inhibits Human Breast Cancer Progression in Preclinical Models"

_nutrients, 2024, doi:10.3390/nu16142369_

Round 1
Reviewer 1 Report
Comments and Suggestions for Authors
This article mainly describes the inhibitory effect of manuka honey on breast cancer MCF-7, and makes a certain theoretical basis for the application of manuka honey. However, there are still many problems in the article should be revised and reevaluate, as follows.
1. Manuka honey (MH) as a mixture, the article did not specify the specific anti-tumor effect of which component? Whether each component plays a role, or different components play a different role, which component plays the main role, which component plays an auxiliary role, need to be further studied.
2. In line 18-19, why does MH have a good inhibitory effect on MCF-7 breast cancer cells, but not an ideal effect on highly metastatic MDA-MB-231 cells?
3. In lines 22-23, what is the significance of comparing the range of action of MH to that of anti-estrogenic tamoxifen?
4. In the Fig1A, it is suggested to increase the cytotoxicity test of Manuka honey on HMEC cell to improve the safety evaluation of whether it is safe at 5% (w/v).
5. In vivo data, the frequency of tumor volume recording was different and it is recommended to maintain consistency and unify the image aesthetics. And in the control group, 0-24 days of oral glucose solution can also play a role in inhibiting tumor growth? After 24 days, the tumor suddenly increased in size. In the MH group, the tumor disappeared directly after MH treatment, whether the individual difference was too large. The results were questionable, and it was suggested to increase the scale in the Fig4B.
6. It is recommended to add camptothecin as a positive control in the Fig2B. And please explain why the expression of ACTIN protein is blank in Fig2B.
7. It is suggested to increase the semi-quantitative data of WB. The proteins expression of AKT and AMPK in Fig3 A are not uniform, and the expression of STAT3 and GAPDH proteins in Fig3B are not at the same levels, and the result trend is not clear.
Meanwhile, whether it is appropriate to select 5% MES for positive drugs, it is recommended to add clinical MCF-7 treatment drugs such as tamoxifen or camptothecin as positive controls.If the Western blot data in Figure 3 has three duplicate backups and shows the complete membrane sample, it is recommended to upload it to the supplementary file.
8. It was suggested to increase the scale and HE sections of mouse organ tissues to improve the safety evaluation at this concentration of MH.
Comments on the Quality of English LanguageEnglish Language is good but attention should be paid to formatting problems. For the bold problem in the notes, the labels (A) in Fig 1-3 are all different. Moreover, the description of positive drugs and antibodies is more complicated, so it is recommended to simplify the expression.304 lines with an extra period; The different heights of the A and B letters in Figure 2, etc., still need to be carefully examined and modified.
Author Response
"Please see the attachment."

Reviewer 2 Report
Comments and Suggestions for Authors
This article by Márquez-Garbán et al. explores the use of manuka honey in controlling breast cancer growth in vitro and in vivo using breast cancer cell lines. The article is well written and the experimental results are well described. I have 2 minor comments:
1. Why was there no cleaved PARP at 0.6 and 2.5% MH? 2.5% MH treatment shows high % of apoptosis at 48h. Can the authors provide some explanation for this?
2. Why are the tumor sizes so varied in the control arm of the PDX tumor experiment?
Author Response
"Please see the attachment."
